# A Deep Dive into VDAC1 Conformational Diversity Using All-Atom Simulations Provides New Insights into the Structural Origin of the Closed States

**DOI:** 10.3390/ijms23031175

**Published:** 2022-01-21

**Authors:** Jordane Preto, Hubert Gorny, Isabelle Krimm

**Affiliations:** Centre de Recherche en Cancérologie de Lyon, Université Claude Bernard Lyon 1, INSERM 1052, CNRS 5286, 69008 Lyon, France; hubert.gorny@etu.univ-lyon1.fr (H.G.); isabelle.krimm@univ-lyon1.fr (I.K.)

**Keywords:** voltage-dependent anion channel, beta-barrel transporter, molecular dynamics

## Abstract

The voltage-dependent anion channel 1 (VDAC1) is a crucial mitochondrial transporter that controls the flow of ions and respiratory metabolites entering or exiting mitochondria. As a voltage-gated channel, VDAC1 can switch between a high-conducting “open” state and a low-conducting “closed” state emerging at high transmembrane (TM) potentials. Although cell homeostasis depends on channel gating to regulate the transport of ions and metabolites, structural hallmarks characterizing the closed states remain unknown. Here, we performed microsecond accelerated molecular dynamics to highlight a vast region of VDAC1 conformational landscape accessible at typical voltages known to promote closure. Conformers exhibiting durable subconducting properties inherent to closed states were identified. In all cases, the low conductance was due to the particular positioning of an unfolded part of the N-terminus, which obstructed the channel pore. While the N-terminal tail was found to be sensitive to voltage orientation, our models suggest that stable low-conducting states of VDAC1 predominantly take place from disordered events and do not result from the displacement of a voltage sensor or a significant change in the pore. In addition, our results were consistent with conductance jumps observed experimentally and corroborated a recent study describing entropy as a key factor for VDAC gating.

## 1. Introduction

It is now widely accepted that mitochondria are more than the energy powerhouse of the cell given their importance in cell signaling events, aging, cell proliferation and apoptosis [1,2]. Mitochondrion’s cross-talk with the rest of the cell is directly sustained by the voltage-dependent anion channel 1 (VDAC1)—a β-barrel membrane protein located within the mitochondrial outer membrane (MOM) [3], which regulates the transport of ions [4,5] and metabolites (ATP, NADH…) diffusing in or out of mitochondria [6,7]. VDAC1 plays a central role in many cellular processes ranging from energy metabolism to the triggering of mitochondria-mediated apoptosis [8,9]. For these reasons, VDAC1 is often regarded as a promising target for the treatment of metabolic disorders [10] as well as cancer [11,12] and neurogenerative diseases [13,14] where pro- and anti-apoptotic action are required, respectively. At the molecular level, VDAC1 regulation is characterized by the ability of the channel to switch or “gate” between a high-conducting open state, which is permeable to most ions and metabolites, and multiple low-conducting closed states, which display a reduced ion conductance and are impermeable to large metabolites [6,15]. In normal physiological conditions, VDAC1 is stable in its open state. However, lowering the pH [16] or setting the transmembrane (TM) potential below or above a certain threshold value (below −30 mV and above +30 mV, typically) contributes to the emergence of closed states, which are usually short-lived over experimental timescales [17]. Finally, the open state of VDAC1 was reported to be essentially anion-selective, displaying a 2:1 Cl^−^:K^+^ permeability ratio upon a 0.1–1 M KCl concentration gradient, while the closed states showed a preference for cations [15].

Several experimental structures of VDAC have been obtained so far using various techniques, including X-ray crystallography [18,19], NMR [20,21,22] and Cryo-EM [23]. Overall, the channel was depicted as a 19 β-stranded barrel pore, and an α-helical N-terminus positioned halfway inside the barrel (Figure 1). Despite slight differences in the conformation of the N-terminus, all available structures were shown to correspond to the open state of the channel, while no structural description of the closed states or of the gating mechanism has been validated [24,25]. As the impact of channel gating on energy metabolism or on the induction of apoptosis is still uncertain [26], structural characterization of the closed states turns out to be crucial to provide a molecular basis for VDAC1 functioning and to better understand its role at the cellular level. In addition, confirmed closed conformers might constitute valuable docking targets in order to develop new drugs which activate or inhibit channel gating in various ways depending on the pathology.

Over the past decade, many attempts have been made to highlight the structural transition(s) at the basis of VDAC gating [27,28,29]. Ujwal et al. [18] proposed that closed states originated from a rigid displacement of the N-terminal helix toward the center of the pore, thereby reducing the flux of ions and metabolites. Using solid-state NMR spectroscopy and molecular dynamics (MD), Zachariae et al. [28] observed that removal of the N-terminal segment led to a partial collapse of the pore and that high ellipticity could induce both a low conductance and a reversed selectivity characteristic of closed states. As a result, channel gating might be explained by the detachment/translocation of the N-terminus out of the barrel, favoring the so-called semi-collapsed states. More recently, using single-channel electrophysiological experiments [21], it was demonstrated that specific crosslinking between the N-terminus and the β-barrel wall resulted in destabilizing the closed states, implying that the long N-terminal helix is involved in the gating mechanism. Although the above results may sound contradictory at first glance, they support the idea that the VDAC N-terminus plays a central, either direct or indirect, role in the establishment of the closed states.

In a recent publication, we demonstrated that the N-terminus of VDAC1 was an intrinsically disordered region (IDR) [30]. This was shown by comparing published NMR data and molecular simulations performed with an IDR-specific force field. Specifically, we observed that the α-helical content of the N-terminus in the open state was mainly stabilized by interactions with the barrel wall, suggesting that detachment of the N-terminus from the pore would result in the subsequent unfolding of the helical structure. At the same time, simulations carried out on a double-cysteine mutant indicated that the disordered properties of the N-terminus were crucial to generating subconducting states prevailing over hundreds of nanoseconds. To validate this assumption and identify possible structures representing closed states, an in-depth investigation of the conformational space of VDAC1 WT was performed in the present work. First, accelerated MD (aMD) simulations were run at applied voltages using the IDR-specific force field validated in our previous study. A total of 2 µs per voltage, based on multiple trajectories initiated from the open state, was used to maximize the exploration of the landscape and to generate large backbone changes of the channel. The voltage was set to ± 40 mV, which lies within the range where closed states become observable. Next, unbiased MD was run to identify conformers with durable low-conducting properties. We found that the only structures exhibiting a relatively stable low conductance compatible with closed states were characterized by a specific positioning of the unstructured N-terminal tail, which partially occluded the pore. While the overall position of the N-terminal tail was sensitive to the applied voltage, highly disordered dynamics of the unfolded N-terminus indicated that random events made a significant contribution to the establishment of low-conducting structures. Moreover, subconducting conformers did not result from the rigid displacement of a voltage-sensing domain [31] or a significant change in the ellipticity of the channel, as previously suggested [28]. The validity of the generated models is supported by discontinuous adjustment of the conductance when the channel transitions back to its open state, which is in line with conductance jumps measured in electrophysiology [32]. Finally, we should stress that our low-conducting structures displayed no cation selectivity, in contrast to experimental closed states. Nevertheless, recent studies on VDAC1 have suggested that low conductance and cationic selectivity may have a different structural origin [33], meaning that our sampled conformers still constitute reasonable models to explain the low conductivity of the closed states.

## 2. Results

### 2.1. aMD Simulations

To investigate the conformational changes experienced by VDAC1 at applied voltage, exploratory aMD simulations were conducted starting from the 3EMN structure (Figure 1), which is known to correspond to the open state of the channel [18]. aMD was run at +40 mV and −40 mV, where the voltage was set by applying a constant electric field on the embedded channel. Due to the intrinsically disordered properties of the N-terminus, the IDR-specific force field ff14IDPSFF [34], which was shown to accurately reproduce NMR shifts in the N-terminal region [30], was applied. Each aMD simulation was conducted over a total of 2 μs, based on multiple trajectories (walkers) to enhance the exploration process (Section 4.3). In Figure 2, we have depicted the conformational space generated at both +40 mV and −40 mV in terms of the first two principal components (PCs) computed from the cartesian coordinates of the N-terminal atoms. As a comparison, we also showed the region explored over 500 ns of unbiased MD at 0 mV where no modification of the structure of VDAC1 was reported. Importantly, no major conformational changes were found either by running unbiased MD at +40 mV or −40 mV (not shown), which justifies the use of biased simulations for the exploration of the channel.

As seen in Figure 2, the most important structural variations observed from aMD are related to the secondary structure and the location of N-terminal residues M1 to G11, which appear to correlate with the first two PC coordinates. Specifically, the 3_10_-helix, which involves residues Y7, A8 and D9 in the crystal structure, was found to be mostly disordered in our sampled conformers. At both +40 and −40 mV, all the residues up to G11 were able to detach from the barrel and unfold, with a tendency to move toward positive potentials, i.e., upward at +40 mV and downward at −40 mV. This result was supported by the little conformational overlap found between the two voltages (dashed green contour in Figure 2), which implies that the dynamics of the M1-G11 tail were sensitive to the applied voltage and not only driven by random events. Conformers with high PC1 were characterized by the unstructured M1-G11 segment positioned higher in the pore (Figure 2b–d), while this segment was located in the bottom part of the barrel at low PC1 values occasionally coming out of the lumen from below (Figure 2e–g). Unlike the 3_10_-helix, the long α-helix involving residues K12 to T19 in the open state was more conserved throughout our exploratory simulations, although unfolding was also reported for residues located at both ends of the helix (Figure 2e–h). Similar observations about the stability of the N-terminal helices were made on the basis of the secondary-structure propensity measured at each voltage (Appendix A), which indicated that the 3_10_-helix present in the native VDAC1 structure is less stable and more subject to important conformational variations as compared with the long helix.

To get a better understanding of the impact of the voltage on the changes observed in the N-terminal backbone, we have recorded the position of N-terminal charged residues, namely, D9, K12, R15, D16 and K20, over the course of aMD simulations. Average positions of Cα atoms along *z*, i.e., the direction of the field, are shown in Figure 3. As a comparison, the *z*-coordinate of these atoms was also provided for the 3EMN structure. Although standard deviations were found to be significant for several residues (which was expected due to the disordered state of the N-terminus), residues D9 and K20 seemed more sensitive to the voltage orientation than other charged residues. More precisely, D9, a negatively-charged residue, showed a certain tendency to move upward at +40 mV and downward at −40 mV, with respect to the crystal structure. Similar sensitivity was observed for K20, a positively-charged residue, moving in the opposite direction as D9. Conversely, we realized that the average positions of residues K12, R15 and D16 were much lower at both +40 mV and −40 mV than in the crystal structure, which indicated that the most probable positional change of the long α-helix was to shift down along the barrel, irrespective of the voltage value. Importantly, the apparent charge of the N-terminal tail M1-G11 was −1.69 at neutral pH and at 150-mM KCl concentration (−1.15 for the N-terminal “tip” M1-P5), whereas the apparent charge of the K12-T19 segment was +0.07 in the same conditions, which is consistent with the present results showing that the N-terminal tail is more influenced by the voltage, as compared with the K12-T19 helix.

### 2.2. Investigating Subconducting States of mVDAC1

Out of all the conformers sampled in aMD, we wanted to identify those exhibiting a low conductance, as is the case of VDAC1 closed states. VDAC1 closed states were experimentally reported to have a conductance of about half the value in the open state irrespective of the ion concentration [15]. In our study, the conductance was estimated from unbiased, i.e., standard, MD simulations by recording the number of ions crossing the channel at applied voltage. While exploratory aMD simulations were performed at a physiological, i.e., 150 mM, KCl concentration, conductance evaluation was conducted at 1M KCl in order to enhance the number of ion crossing events and, in turn, improve the accuracy of the predictions. By keeping the system unrestrained, MD over other techniques, like Grand-Canonical Monte-Carlo Brownian Dynamics (GCMC/BD) [35], offers the advantage of testing the stability of the selected structure over time. In this way, transient vs. stable low-conducting conformers can be identified.

As a control, we first ran a 500 ns long unbiased MD trajectory at +40 mV starting from the 3EMN crystal structure. During the simulation, 238 K^+^ and 529 Cl^−^ ions were found to pass completely through the channel, going downward and upward along the pore axis, respectively. The net electric current was deduced from the slope of the total number of crossing charges as a function of time, whereby a simple linear regression gave an R^2^ of 0.997. As the diffusion of ions is usually overestimated in MD, the bulk diffusion coefficient of potassium chloride was computed separately from simulation of a low-concentrated KCl solution and compared with the experimental bulk diffusion coefficient at the same concentration (see Section 4.5). As suggested in [25], the ratio of the two values was used to renormalize the electric current. After renormalization, the current was estimated at 148.43 pA, which is equivalent to a conductance of σ=148.43 pA /40 mV=3.71 nS and was consistent with the conductance of the open state measured experimentally, e.g., 3.7 nS in [18] and 4.0 nS in [32]. Moreover, the ratio between chloride and potassium permeation events, also called the current ratio, was 2.2 at the end of our simulation, which agreed with previous computational studies on the open state [24].

As mentioned above, unbiased MD was used next to evaluate the conductance and the stability of the structures generated in aMD. As a lot of conformers were sampled over the course of our simulations, namely, 20,000 conformers at +40 mV and 20,000 at −40 mV, RMSD-based clustering analysis was first performed to extract 200 diverse structures of the channel at each voltage. GCMC/BD was then run on these structures to provide a rough estimate of their conductance at 1 M KCl and select those to be further checked from unbiased MD. Specifically, we selected conformers for which GCMC/BD predicted a conductance less than 2.5 nS, which corresponds to 67% of the conductance in the open state (here taken as 3.7 nS). In total, selected frames included 55 out of 200 extracted from aMD at +40 mV and 16 out of 200 extracted from aMD at −40 mV. GCMC/BD results are summarized in Appendix A.

By running unbiased MD, we identified a total of 3 conformers whose trajectory was characterized by a stable low conductance over 500 ns (σ≤2.3 nS and $R^{2}\geq 0.99$). These included two conformers sampled at +40 mV, referred to as A and B, and one sampled at −40 mV, referred to as C. In Figure 4a, we showed the total number of ion permeation events recorded for these structures, as well as for 3EMN used here as a reference. To be consistent with our original sampling, the voltage was set to +40 mV for conformers initially generated at +40 mV and to −40 mV for conformers generated at −40 mV. Estimated conductances are displayed in Figure 4b. Importantly, conductances of conformers A, B and C were in line with experimental values for closed states, which are expected to lie within an interval of 40–60% of the conductance in the open state [36]. Finally, the average (representative) structure of each MD trajectory was depicted in Figure 4c. Structures of subconducting trajectories all featured an unfolded region of the N-terminus obstructing the middle of the pore in the *xy*-plane, i.e., the plane parallel to the membrane. Important differences were also observed between the +40 mV and −40 mV cases: at +40 mV, the unfolded domain corresponded to the M1-G11 segment, which passed through the center of the barrel along the *z*-axis, i.e., the pore axis. Meanwhile, at −40 mV, the unfolded segment also included residues up to A14 and essentially occluded the bottom part of the barrel along the *z*-axis.

Given the average structure of each low-conducting trajectory as compared with 3EMN, the specific position of the unfolded N-terminal region is likely to explain the low conductance. This statement is supported by unbiased MD simulations performed on additional conformers extracted from aMD at +40 mV, which exhibited an unstable low conductance (Figure 5). Such conformers displayed two conducting regimes: a low-conducting regime where the conductance was about half the conductance in the open state and a high conducting phase with a conductance equaling that of the open state. Closer inspection of the trajectories revealed that bifurcation specifically occurred when the unfolded tail, which had a relatively stable yet metastable horizontal position in the middle of the pore, moved back to a vertical position along the barrel, thus re-opening the pore (Figure 5c). Importantly, the conductance was found to switch abruptly from a low to high value with no values in between, which is in line with electrophysiological experiments where the same observations are made for closed-to-open transitions [32].

In addition to the above, a deep analysis of our stable subconducting trajectories A, B and C was carried out. Frames taken at regular time intervals over the course of each trajectory are displayed in Appendix A. Although the low conductance characterizing A and B was attributed to the specific positioning of the N-terminal tip M1-P5 in the middle of the pore, MD trajectories indicated no recurring interactions between the segment and the β-barrel, making the former jiggle upon thermal noise fluctuations. The apparent stability of A and B in a low conductance state was expected to be due to electrostatic interactions happening between N-terminal residues located upstream in the sequence (T6 to G11) and the barrel wall. Long-lived interactions including ionic, aromatic and hydrogen bonds were observed between the two regions resulting in the formation of a stable “hooked” T6-G11 segment, which was presumed to maintain the unfolded tip in a horizontal position in the middle of the channel (Appendix A). In contrast to A and B, the M1-P5 tip remained globally attached to the barrel wall in trajectory C. This time, partial occlusion of the pore directly involved the unfolded T6-G11 segment, which was detached from the β-barrel and remained in the middle of the pore in the xy-plane (Appendix A). Further analysis revealed recurrent salt bridges and hydrogen bonds at P5-N125 and K12-G203, which were expected to be responsible for the stable position of the T6-G11 segment.

As discussed in a previous study [28], we also investigated whether the ellipticity of the barrel was responsible for the low conductance of our conformers. To that purpose, we computed the ellipticity of the barrel over the course of our subconducting trajectories and compared the results with the ellipticity recorded for the 3EMN trajectory (Appendix A). While the ellipticity was generally higher for low-conducting states, we also observed long time intervals during which ellipticity was comparable to that of 3EMN (e.g., B and C between 350 ns and 500 ns in Appendix A). As the conductance was stable all along our trajectories, this shows that the measured low-conducting properties were not correlated with the ellipticity of the channel.

Finally, we also recorded, for each subconducting trajectory, the ratio of Cl^−^ over K^+^ crossing events (also called current ratio, i.e., $I_{Cl-}$/$I_{K+}$), which provides an indication of channel selectivity. Current ratios are provided in Table 1 for MD trajectories initiated from 3EMN, as well as from conformers A to E. Notably, unstable subconducting states D and E, after the bifurcation point, displayed a current ratio similar to 3EMN, confirming that the channel transitioned back to the open state. In all of the other cases, which correspond to subconducting states, the ratio was higher than the 3EMN ratio, which reflects an increase in anion selectivity. As VDAC1 closed states were shown to be cation-selective [37], this implies that additional structural changes might be needed to fully account for closed states. Possible strategies on how to describe cation selectivity, which, in the case of VDAC1, might be uncorrelated with a decrease in the conductance value [33], are discussed in Section 3.

### 2.3. Refolding of the N-Terminus

In search of other possible structures explaining VDAC closed states, we also identified a few aMD trajectories whereby the N-terminal tail was able to unfold and refold into a helical pattern different from that of the crystal structure. This was observed for two trajectories: the first one (trajectory 1) was extracted from aMD at +40 mV (conformational landscape provided in Figure 2), while the second one (trajectory 2) was obtained from short additional aMD using a stronger biasing potential to guarantee complete unfolding of the N-terminus. Both trajectories are displayed in Figure 6, starting from the 3EMN crystal structure and ending at the conformer with refolded N-terminus. Trajectory 1 was characterized by the refolding of the Y7-D9 3_10_-helix into a long α-helix, including residues L10 to K12. Such a transition goes along with the loss of helical structure for residues K12 to T19. Notably, the refolded structure had residues Y7 to D9 positioned similarly to the crystal structure (RMSD = 0.66 Å), which also explained the PC coordinates used to plot Figure 2, which account for large movements of the N-terminus, could not properly highlight this new state. Alternatively, Trajectory 2 involved the detachment of residues Y7-R15 from the barrel and subsequent unfolding of the long helix. Subsequently, residues Y7 to D9 were found to refold into a new α-helical structure via the interaction with barrel residues located on the wall opposite to where the N-terminus was initially positioned (H-bonds with E59, R63 and K96).

The stability and conducting properties of the refolded structures, i.e., the final structure of each trajectory displayed in Figure 6, were tested by performing unbiased MD at +40 mV. As shown in Appendix A, we found that the refolded state from trajectory 1 was stable over 500 ns and displayed a conductance similar to that of the open state. On the contrary, the new helical state in trajectory 2 was stable up to 270 ns and showed a conductance reduced by half, compatible with closed states. After 270 ns, the new helical N-terminal domain was able to detach again from the barrel and to return to a conformation with a high conductance similar to the open state.

## 3. Discussion

Despite the central role played by the mitochondrial transporter VDAC1 in cell homeostasis and apoptosis [8,9], a molecular description of its functioning, particularly of its gating mechanism, is still missing. Specifically, although a few structures of VDAC open state have been reported [18,19,20,21], the structural profiles of closed states, which take place at high enough (in magnitude) TM potentials, have yet to be determined [24,25]. Therefore, in an attempt to identify conformations responsible for the VDAC1 closed states and to investigate open-to-closed transitions, we have conducted an extensive MD-based study following a two-step strategy.

First, a broad exploration of the conformational landscape of VDAC1 was performed at applied voltages using aMD, a biasing technique shown to be successful in simulating protein transitions normally accessible on millisecond timescales. To enhance the sampling process, aMD was run at +40 mV and −40 mV based on multiple trajectories over a total of 2 μs per voltage. This step generated many conformers characterized by various positions of its N-terminal domain which is expected to play a major role in VDAC gating. Using aMD, we observed that the N-terminus was able to detach from the β-barrel pore and partially unfold. While a large ensemble of conformers was sampled at both +40 mV and −40 mV, representative of a highly disordered state, the dynamics of the unfolded M1-G11 tail were found to be partly driven by the voltage, as the segment showed a tendency to diffuse toward positive voltage values. This result is consistent with the apparent negative charge recorded in this region (−1.69 and −1.15 for the M1-G11 and the M1-P5 segments, respectively), indicating that positive vs. negative voltages would affect the N-terminal tail in opposite ways. On the contrary, conformational changes involving the α-helical segment K12-T19 (apparent charge of +0.07) turned out to be independent of voltage orientation. The main changes included a shift of the helix down the barrel, irrespective of the voltage value. In a previous study [36], it was proposed that the K12-R15-K20 motif played a key role in VDAC1 charge transport and possibly in gating. Given the close proximity of three positively-charged residues, the KRK motif might indeed contribute as a possible voltage sensor for VDAC1. However, the apparent neutral charge of the K12-T19 helix—which also involves the negatively-charged residue D16—as well as the absence of large conformational response due to the voltage in the present study, indicated that the KRK motif is unlikely to behave as a sensing domain. Although the average position of the M1-G11 tail was found to be influenced by the voltage, further investigation would be needed to illuminate the real impact of the TM potential on VDAC1 dynamics, e.g., on the detachment/unfolding of the N-terminus. Methods for recovering the equilibrium distribution from out-of-equilibrium data (like our aMD samples) can be used to estimate the free-energy landscape of the channel and explore how critical energy barriers vary depending on the voltage. Such methods, including the Weighted Histogram Analysis Method (WHAM) [38] and Markov State Models (MSMs) [39], could be employed in the future to investigate VDAC1 equilibrium properties at various voltage values.

As a second step, we investigated the conducting properties of the conformers sampled in aMD to identify structures representing stable subconducting states, as is the case of VDAC closed states. By running 500 ns long MD trajectories, we identified a total of three conformers with stable low conductances—two from aMD at +40 mV and one from aMD at −40 mV—consistent with the conductance reported for closed states (Figure 4) [18,32]. Subconducting structures were characterized by the unfolded N-terminal tail M1-G11 (or even M1-A14 at −40 mV) rearranged in a specific way in the middle of the pore, leading to steric hindrance of the channel. Using MD, we also noticed a few aMD conformers showing a spontaneous loss in their subconducting properties, switching suddenly from approximately half to 100% of the open state conductance. This occurred upon unprompted repositioning of the N-terminal tail M1-G11 starting from a horizontal position in the middle of the pore to a vertical position along the barrel. Importantly, such abrupt changes in the conductivity are routinely observed in electrophysiological experiments where gating transitions take place in a non-continuous way, displaying no intermediate conductance values [32]. In addition, we would like to point out a recent electrophysiological study showing that VDAC1 gating is either based on an entropy-driven mechanism or on enthalpy-entropy compensation [40]. This is in agreement with the current study, where a highly disordered state was generated from the detachment/unfolding of the N-terminus tail, whereas transitions to stable subconducting conformers occurred from specific rearrangement of the unstructured segment, in which random thermal fluctuations clearly play an important role. For these reasons, we believe that the subconducting states identified in this study constitute a robust model to account for VDAC1 closed states.

We should mention that our subconducting conformers exhibit no cation selectivity in contrast to experimental closed states [7,15]. A possible reason for this is that the standard MD techniques used in the present work do not explicitly account for electronic charge redistribution. In cases of a high-amplitude voltage or pH variation that leads to a change in the electrostatic profile of VDAC1, important modifications in residue ${\mathrm{pK}}_{a}$s may be critical to induce cation preference upon gating. This is consistent with several mutagenesis studies, revealing that specific mutations on VDAC1, including K20E or K61E, were sufficient to modify channel selectivity [41,42]. At the same time, previous experimental works have pointed out the existence of a cationic open state for VDAC1 WT [43] together with the lack of obvious correlation between PcVDAC conductance and selectivity [33]. Therefore, it is tempting to think that changes in conductance and selectivity observed during gating have a different structural origin, which would explain why low-conducting structures with anion preference were observed in the present study. Assuming that a reversal in channel selectivity is actually induced by a redistribution of electronic charges, computational validation might be challenging as standard MD techniques and force fields—including those used in the present work—usually involve fixed charges for each atom, which are unaffected by the presence of an external field. In this case, we believe polarizable force fields [44], as well as constant pH MD [45], constitute valuable tools to account for such a mechanism. Finally, we highlighted that the N-terminal domain, as an IDR, was capable of unfolding and refolding into new helical motifs upon interaction with different regions of the barrel. For instance, one of our refolded states featured a helix made of residues Y7 to D9, which interacted with the barrel wall opposite to where the N-terminus was initially attached. As mentioned in [46], we believe that the ability of the N-terminus to refold could provide an explanation on why VDAC1 is able to interact with so many partners, including BCl2 family members [47] or Hexokinase (HK) [48]. Indeed, induced folding, via the formation of new structural patterns, might ensure that the N-terminal domain fits into binding cavities of various structural shapes. However, further work to experimentally characterize some of the many protein complexes involving VDAC1 is needed to confirm or disprove the above assumption.

## 4. Materials and Methods

### 4.1. System Preparation

aMD exploratory simulations were all initiated from the crystallographic structure of mVDAC1 [18] (PDBID: 3EMN, resolution 2.3 Å). The structure was protonated at neutral pH using the propKa algorithm available in the MOE 2018 software (Chemical Computing Group ULC, Montreal, QC, Canada) [49]. Importantly, the membrane-facing residue E73 was left in its unprotonated form as discussed and reviewed in [36]. The channel was embedded in a lipid bilayer using the CHARMM GUI Membrane Builder web server [50] available at: https://www.charmm-gui.org/?doc=input/membrane.bilayer (accessed on 12 September 2020). Prior to embedding, the mVDAC1 structure was centered at the origin and reoriented so that the channel pore was aligned with the *z*-axis [51]. Membrane lipids included dioleoylphosphatidylcholine (DOPC) and dioleoylphosphatidylethanolamine (DOPE) following a 2:1 PC:PE ratio in order to mimic the lipid composition of the MOM [52]. In total, the membrane was made of 186 lipid molecules. The embedded system was set into a box with dimensions 91.9 × 91.7 × 90.2 Å containing around 12,500 TIP3P water molecules at a 150 mM-KCl concentration. As mVDAC1 exhibits a net positive charge of ~3 at neutral pH, the exact number of ions was slightly adjusted in order to neutralize the system.

### 4.2. MD Software and Force Field

The Amber16 package (University of California, San Francisco, USA) [53] was used for aMD and MD simulations. To account for the intrinsically disordered properties of the VDAC1 N-terminus, the IDR-specific force field ff14IDPSFF [34] was applied to residues M1 to G25 while the standard ff14SB force field [54] was used on residues of the β-barrel (L26 to A283). The lipid14 force field [55] was utilized for the lipids of the membrane. Solvent and ions were first minimized by restraining the protein and lipid atoms using cartesian restraints with a spring constant of 10.0 kcal/mol/Å^2^. This was done in two stages, including 5000 steps of steepest descent followed by 5000 steps of the conjugate gradient. The same procedure was repeated one more time without the restraints. Next, the system was progressively heated from 0 to 298 K during 500 ps while restraining heavy atoms of the channel. Finally, simulation in NPT ensemble at a pressure of 1 bar and a temperature of 298 K was run without restraints during 10 ns to complete the equilibration process.

### 4.3. aMD Simulations

Exploration of the conformational space of mVDAC1 at TM potentials of +40 mV and −40 mV was carried out using accelerated molecular dynamics (aMD). aMD is an MD-like method where a bias potential is added to the true potential to facilitate the crossing of high-energy barriers [56]. In its dual-boost version, which was used in the present study, the aMD potential ΔV(x) depends on both the total potential energy of the system V(x) and on its dihedral energy $V_{d}\left(x\right)$ such that:$$\Delta V\left(x\right)=\frac{{\left(E_{p}-V\left(x\right)\right)}^{2}}{\alpha_{p}+E_{p}-V\left(x\right)}+\frac{{\left(E_{d}-V_{d}\left(x\right)\right)}^{2}}{\alpha_{d}+E_{d}-V_{d}\left(x\right)}$$
here $E_{p}$ and $E_{d}$ are the average potential and dihedral energies of the biased system that were computed from the potential and dihedral energies of the unbiased system $E_{p,0}$ and $E_{d,0}$ such that $E_{p}=E_{p,0}+\gamma_{p}.n_{atoms}$ and $E_{d}=E_{d,0}+\gamma_{d}.n_{residues}+\gamma_{d}\prime .n_{lipids}$, where $n_{atoms}$, $n_{residues}$ and $n_{lipids}$ are the number of atoms (including solvent atoms), residues and lipids, respectively. Approximated values of $E_{p,0}$ and $E_{d,0}$ were obtained from our 10 ns-long equilibration run mentioned in Section 4.2. Other parameters were set to $\gamma_{p}=0.16\;\mathrm{kcal}/\mathrm{mol}/\mathrm{atom}$, $\gamma_{d}=3.5\;\mathrm{kcal}/\mathrm{mol}/\mathrm{residue}$ and $\gamma_{d}^{\prime }=30.0\;\mathrm{kcal}/\mathrm{mol}/\mathrm{lipid}.$ To speed up the exploration process and prevent the system from being trapped in local energy minima, the following strategy was used at each applied voltage. A total of 20 aMD trajectories (walkers) were run simultaneously. After 5 ns, all the trajectories were stopped. Clustering analysis based on the RMSD of N-terminal atoms was conducted in order to generate 20 clusters using all the conformers generated so far. Next, the representative structure of each cluster was used as a starting point for a new 5 ns-long aMD trajectory, resulting in 20 new aMD trajectories running simultaneously. After 5 ns, all the trajectories were stopped, and so on. The above procedure was repeated 20 times, which corresponds to a simulation time of 20×20×5 ns=2 μs at each voltage. Notably, the TM potential was set by introducing a constant electric field across the entire simulation box and perpendicular to the membrane (using “efz” and “efn” options in Amber16). The electric field E was calculated so that L.E, where L is the length of the simulation box, equals the desired voltage values, i.e., +40 mV or −40 mV. In all our trajectories, the structure of the channel was saved every 0.1 ns, resulting in 2000/0.1=20,000 frames saved at each voltage.

### 4.4. PB/PNP and GCMC/BD

RMSD-based clustering analysis was run on aMD conformers in order to extract 200 representative frames (out of 20,000) at both +40 mV and −40 mV. RMSD was computed from all the non-hydrogen atoms of the channel, including N-terminal and barrel atoms. At this stage, Poisson-Boltzmann and Poisson-Nernst-Planck (PB/PNP) and Grand Canonical Monte Carlo/Brownian Dynamics (GCMC/BD) were performed on each of the 200 structures to provide a quick estimate of their conductance. PB/PNP and GCMC/BD were run using the open-source programs of the same name, available at http://www.charmm-gui.org/?doc=input/gcmcbd (accessed on 12 September 2020). We refer the reader to [35,57] for more detail about both methods. Input files for PB/PNP and GCMC/BD were automatically built using the CHARMM-GUI Ion Simulator webserver (accessed on 19 July 2021) [58]. During this step, each structure was re-embedded in a 35-Å-thick membrane oriented along the z-axis and positioned in the middle of an $85\times 85\times 95\;{\text{Å}}^{3}$ box. The thickness of buffer regions at the top and bottom of the simulation box were set to 5 Å, while a 1 M KCl concentration was applied on both sides of the membrane. First, PB/PNP was run to estimate input parameters for GCMC/BD calculations; this includes the determination of the protein electrostatic and steric potentials carried out at each point of a rectangular grid with a grid spacing of 0.5 Å. Then, GCMC/BD was run on each frame (1 run per frame) by performing 1 GCMC step for every BD step, for a total of $2\times 10^{7}$ BD cycles. A time step of 0.01 ps was used, leading to 200 ns-long GCMC/BD trajectories. To ensure consistency with our original sampling, TM potentials of +40 mV and −40 mV were applied for conformers initially sampled at +40 mV and −40 mV, respectively. Conductance was estimated by summing up K^+^ and Cl^−^ currents and dividing by the applied voltage. At both voltages, frames with a conductance less than 2.5 nS, which corresponds to 67% of the conductance measured in the open state, were selected as a starting point for unbiased MD in order to check their stability as low-conducting structures.

### 4.5. Unbiased MD

To estimate the conductance of the open state as well as that of conformers selected from GCMC/BD, unbiased MD was performed without restraints during 500 ns (1 trajectory per conformer). Simulations were conducted at a KCl concentration of 1 M to enhance permeation events. Since aMD conformers were initially generated at 150 mM of KCl, ion concentration was increased by adding the right number of ions and by equilibrating the system during 1 ns in the NPT ensemble before production. Just like GCMC/BD trajectories, the applied voltage was set to +40 mV for aMD conformers sampled at +40 mV and to −40 mV for conformers generated at −40 mV. For the equilibrated 3EMN structure, which corresponds to the open state, the voltage was arbitrarily set to +40 mV. As in aMD, the applied voltage was introduced via a constant electric field (see Section 4.3). For each trajectory, crossing events were assumed to occur whenever an ion crossed the center of mass of the channel along the pore axis. To estimate the electric current, linear regression was performed to estimate the slope of the total number of crossing charges $q_{tot}$ vs. time. Here, $q_{tot}$ is given by $q_{tot}\left(t\right)=\left(+1\right)n_{K+}\left(t\right)+\left(-1\right)n_{cl-}\left(t\right)$, where $n_{K+}\left(t\right)$ and $n_{cl-}\left(t\right)$ are the number of K^+^ and Cl^−^ crossing events recorded up to time t, which take into account the direction in which each ion is crossing the channel. In theory, the current should be directly deduced from the slope of $q_{tot}\left(t\right)$. However, ion diffusion is usually overestimated by MD programs. Therefore, the bulk diffusion coefficient of a KCl solution at a concentration of 10 mM was first estimated by MD simulation ($D_{sim}$) and compared with the experimental value reported at the same concentration ($D_{\exp})$. Values of $D_{sim}$ and $D_{exp}$ were taken from our previous study [30], i.e., $D_{sim}=3.273\;\cdot \;10^{-5}{\mathrm{cm}}^{2}/s\;$and $D_{exp}=1.918\;\cdot \;10^{-5}{\mathrm{cm}}^{2}/$s. The electric current was deduced by rescaling, i.e., by multiplying the slope of $q_{tot}\left(t\right)$ by $D_{exp}/D_{sim}$. Finally, the conductance was approximated by dividing the current by the applied voltage.

### 4.6. Clustering Analysis

Multiple clustering analyses were performed throughout this study using the cpptraj module of AmberTools18 (University of California, San Francisco, CA, USA) [59]. Structures were first superimposed on all the backbone atoms of the channel. Frames were then clustered by using an RMSD-based agglomerative hierarchical algorithm and by requiring a fixed number of clusters as output. Pairwise RMSDs were either determined from the N-terminal atoms only (Section 4.3) or from all the atoms of the channel, including the N-terminal and barrel atoms (Section 4.4). In all cases, hydrogen atoms were excluded from RMSD calculations.

### 4.7. Molecular Visualization

Visualization of generated structures and trajectories was carried out using VMD (University of Illinois at Urbana–Champaign, IL, USA) [60] and MOE2018 [49], where the former was also utilized to draw the structures of the present manuscript.

## 5. Conclusions

The present study aimed to explore the structural origin of the closed states of the voltage-dependent anion channel (VDAC1), which have not been characterized so far. By using all-atom simulations, a large region of the conformational landscape of the channel was explored at typical voltages promoting channel closure. Specifically, we observed that the conformational diversity of VDAC1 at applied voltages was related to its N-terminus, which was able to detach from the barrel pore and to unwind. At this stage, a highly disordered state of the channel characterized by various positions of the unstructured N-terminal domain (M1-A14) was found to take place, interacting with the barrel in various ways. In addition, we observed that the negatively-charged N-terminal tail (M1-G11) was sensitive to voltage orientation while no apparent voltage-sensing domain was identified. Next, conducting properties of our VDAC1 conformers were tested. A few conformers were shown to exhibit a stable low conductance, similar to that of closed states. Low-conducting structures were all characterized by a particular arrangement of the unfolded N-terminal tail, which obstructed the pore in a steric fashion. Although no cation selectivity was observed, our subconducting models were shown to share a number of common features with experimental closed states, thus providing a robust structural basis to account for channel closure. As suggested by other studies, we speculate that cation selectivity has a different structural origin from the low conductance and could result from a change induced by the voltage in the ${\mathrm{pK}}_{a}$s of well-chosen residues.

## Figures and Tables

**Figure 1 ijms-23-01175-f001:**
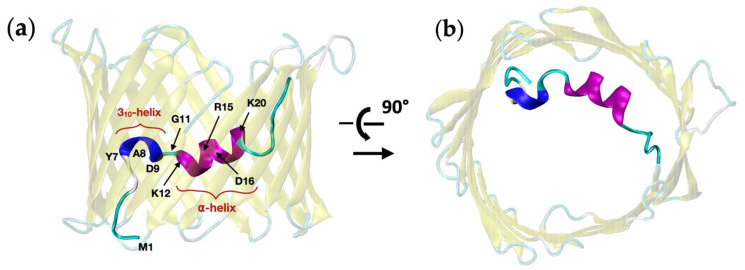
Structure of mVDAC1 solved by crystallography (PDB ID: 3EMN, adapted from Ref. [18], 2.3 Å-resolution) attributed to the open state of the channel. Key residues and major structural components of the N-terminus (residues M1 to G25), including the 3_10_- and alpha helices, are shown. (**a**) Side view (MOM), (**b**) top view.

**Figure 2 ijms-23-01175-f002:**
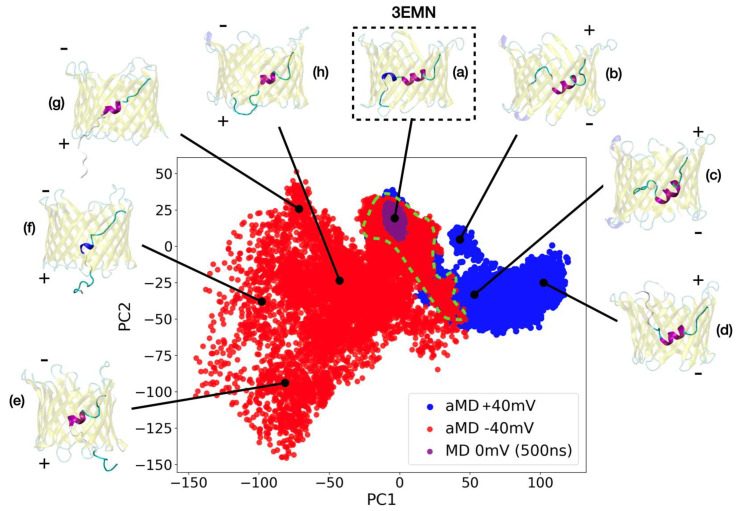
Conformational landscape of mVDAC1 sampled from aMD at +40 mV (blue) and at −40 mV (red) (total simulation time at both voltages was 2 μs). Conformations generated from unbiased MD over 500 ns are also shown (purple). Each conformer is given by its first two principal components computed from N-terminal atoms using all the aMD structures. The area delimited by the dashed green contour corresponds to the region of overlap between the two voltages. Representative structures of the different PC regions are also displayed. (**a**) Stands for the 3EMN crystal structure; (**b**–**d**) correspond to conformers generated at +40 mV while (**e**–**h**) represent conformers generated at −40 mV.

**Figure 3 ijms-23-01175-f003:**
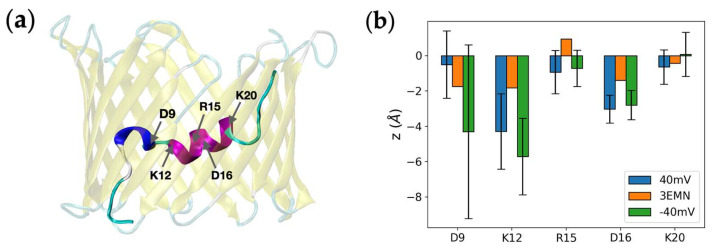
(**a**) Crystal structure of VDAC1 showing charged residues involved in the N-terminus. (**b**) Average position along z of the Cα atom of each charged residue obtained from aMD simulations at +40 mV and −40 mV. z=0  Åroughly corresponds to the middle of the pore. Error bars corresponding to standard deviations were also added.

**Figure 4 ijms-23-01175-f004:**
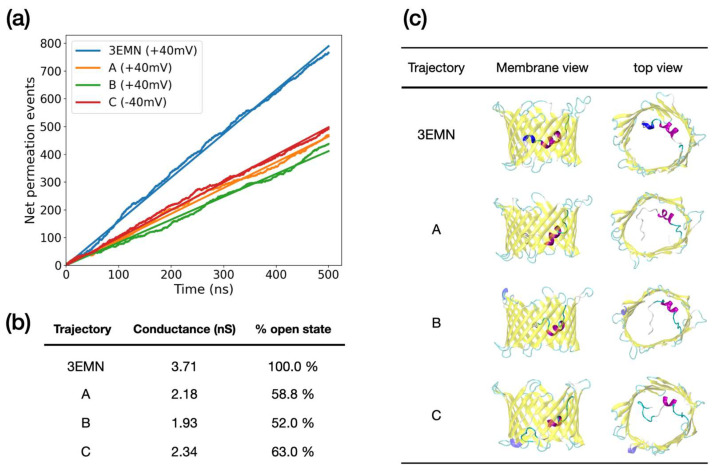
(**a**) Net ion permeation events recorded from unbiased MD simulation at applied voltage. Simulations were initiated from different VDAC1 conformations, namely, 3EMN and 3 conformers obtained from aMD; conformers A and B were sampled at +40 mV and conformer C was sampled at −40 mV. To be consistent with our initial sampling, applied voltage in unbiased MD was set to +40 mV for 3EMN, A and B, and −40 mV for C. For each curve, simple linear regression was performed (straight lines), giving an R^2^ of 0.99 in all cases. (**b**) Conductance deduced from the slope of the ion permeation curves. The slopes correspond to the electric current, which, after renormalization and division by the applied voltage, provide an estimate of the conductance. (**c**) Average structure of MD trajectories in side view (membrane) and top view.

**Figure 5 ijms-23-01175-f005:**
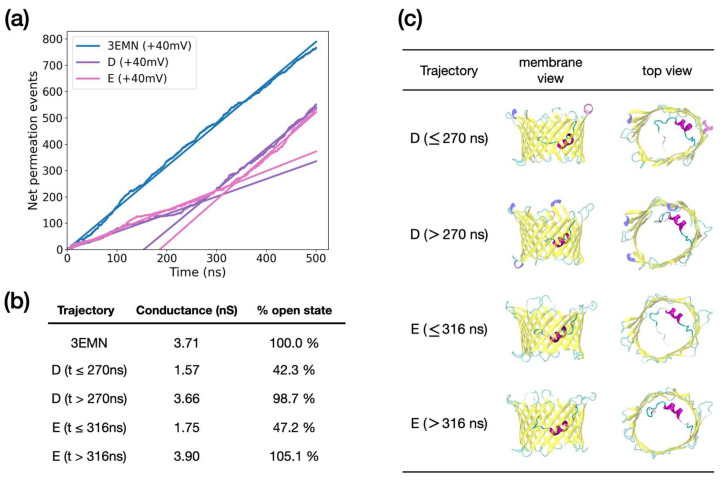
(**a**) Net ion permeation events recorded from unbiased MD at applied voltage. Trajectories were initiated from 3EMN and 2 conformers extracted from aMD at +40 mV, referred to as D and E. For each curve, simple linear regression was performed (straight lines). As trajectories D and E exhibited two conducting trends, two linear regressions, one before and one after the bifurcation point, were conducted. Bifurcation points were estimated at 270 ns and 316 ns for D and E, respectively. (**b**) Conductance deduced from the slope of the ion permeation curves. The slopes correspond to the electric current, which after renormalization and division by the applied voltage, provides an estimate of the conductance. (**c**) Average structures of MD trajectories in side view (membrane) and top view.

**Figure 6 ijms-23-01175-f006:**
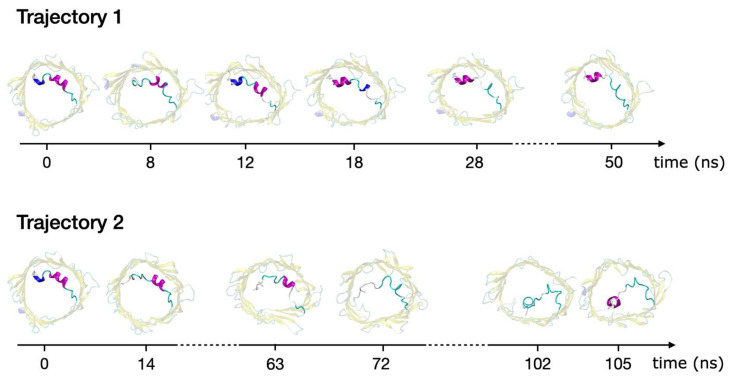
aMD trajectories showing unfolding and refolding of mVDAC1 N-terminal tail. Structures generated all along each trajectory are depicted in top view. The first structure of each trajectory corresponds to a fold similar to the 3EMN crystal structure, while final structures correspond to the refolded N-terminus.

**Table 1 ijms-23-01175-t001:** Conductance and current ratio $I_{Cl-}$ /$I_{K+}$ measured for trajectories started from 3EMN and sampled subconducting conformers. For trajectories D and E, which show two conducting regimes, conductance and current ratio before and after the bifurcation point are provided.

Trajectory	Conductance (nS)	$I_{Cl-}$ $/I_{K+}$
3EMN	3.71	2.21
A	2.18	4.33
B	1.93	3.09
C	2.34	2.80
D (t ≤ 270 ns)	1.57	3.20
D (t > 270 ns)	3.66	2.25
D (t ≤ 316 ns)	1.75	3.27
E (t > 316 ns)	3.90	2.36

## Data Availability

The authors declared the availability of all materials and data upon request.

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
