# Peer review of "A Deep Dive into VDAC1 Conformational Diversity Using All-Atom Simulations Provides New Insights into the Structural Origin of the Closed States"

_ijms, 2022, doi:10.3390/ijms23031175_

Round 1

Reviewer 1 Report

This manuscript investigates closed states of VDAC by molecular dynamics simulations. Both the biased and unbiased simulations performed are quite exhaustive and their analysis quite thorough.

My main and major criticism is that I have significant doubts about their identification of potential VDAC closed states. Indeed, the conformers observed in the authors' trajectories and representing, according to them, closed states resemble the conformers of the NMR structure (PDB code 2K4T) supposed to represent open states (Lee et al. , 2011). Moreover, the authors acknowledge that the conformers they claim to be closed states are not cationic in contradiction with what has been shown for closed VDAC states (Colombini, 1980).

The second point is that the authors describe the conformational space generated at +/- 40 mV in terms of principal components, which relate to the position of the N-terminal segment. However, this being the least constrained it is not surprising that the first two PCs relate to the most mobile segment. The authors should have pushed their PC analysis and excluded the first two PCs to analyse the following ones.

Reviewer 2 Report

In this study, Preto investigated the conformational distribution of VDAC1 and its associated conducting properties with molecular dynamics simulations. Using enhanced sampling, they were able to delineate wide distributions of VDAC1 conformation when voltage is present. Such conformational pools may include the closed state of VDAC1. They then performed the conducting simulation with the selected closed state of VDAC1 obtained from their enhanced sampling simulations. Their results provide insights into the conduction properties of VDAC1 from the structural perspective. I found the manuscript is clearly written, leading to a good story. I therefore support its publication, if the authors can address my following comments:

On page 2, line 80-81. N-terminus of VDAC1 was demonstrated to be an IDP. I am wondering this observation was based on this short segment alone in solution or within the VDAC1? I had this confusion because on next page, line 86-87, “simulations carried out on a double-cysteine mutant indicated that the disordered properties of the N-terminus were crucial to generate subconducting states prevailing over at least hundreds of nanoseconds”. It reads to me that the N-terminus is an IDP also within the VDAC1.

The authors applied aMD simulations to enhance the sampling when the voltage was present and they found a wide distribution of the conformations shown in Figure 2. However, they performed unbiased MD simulations with a short time length and found a narrow distribution without voltage. To me, it is unclear the narrow conformational distribution of VDAC1 without voltage is just simply due to the sampling issue or the intrinsic stable property of VDAC1 itself. To make a direct comparison, the authors should, in my opinion, perform the same simulations with the presence of the voltage.

Figure 3, the average positions should be supplied with error bars when the voltage is present.
